# Towards a Standardised Performance Evaluation Protocol for Cooperative MARL

**Rihab Gorsane**[1][*]      **Omayma Mahjoub**[12][*][†]      **Ruan de Kock**[1][*]      **Roland Dubb**[13][†]

**Siddarth Singh**[1]                          **Arnu Pretorius**[1]

[1]InstaDeep
[2]National School of Computer Science, Tunisia
[3]University of Cape Town, South Africa

## Abstract

Multi-agent reinforcement learning (MARL) has emerged as a useful approach to solving decentralised decision-making problems at scale. Research in the field has been growing steadily with many breakthrough algorithms proposed in recent years. In this work, we take a closer look at this rapid development with a focus on evaluation methodologies employed across a large body of research in cooperative MARL. By conducting a detailed meta-analysis of prior work, spanning 75 papers accepted for publication from 2016 to 2022, we bring to light worrying trends that put into question the true rate of progress. We further consider these trends in a wider context and take inspiration from single-agent RL literature on similar issues with recommendations that remain applicable to MARL. Combining these recommendations, with novel insights from our analysis, we propose a standardised performance evaluation protocol for cooperative MARL. We argue that such a standard protocol, if widely adopted, would greatly improve the validity and credibility of future research, make replication and reproducibility easier, as well as improve the ability of the field to accurately gauge the rate of progress over time by being able to make sound comparisons across different works. Finally, we release our meta-analysis data publicly on our project website for future research on evaluation [3] accompanied by our open-source evaluation tools repository[4].

## 1   Introduction

Empirical evaluation methods in single-agent reinforcement learning (RL)[5] have been closely scrutinised in recent years (Islam et al., 2017; Machado et al., 2017; Henderson, 2018; Zhang et al., 2018; Henderson et al., 2018; Colas et al., 2018a, 2019; Chan et al., 2020; Jordan et al., 2020; Engstrom et al., 2020; Agarwal et al., 2022). In this context, the impact of a lack of rigour and methodological standards has already been observed. Fortunately, just as these issues have arisen and been identified, they have also been accompanied by suggested solutions and recommendations from these works.

---

[*]Equal contribution. Corresponding author: r.gorsane@instadeep.com

[†]Work done during an internship at InstaDeep.

[3]https://sites.google.com/view/marl-standard-protocol

[4]https://github.com/instadeepai/marl-eval

[5]In this paper, we use the term "RL" to exclusively refer to *single-agent* RL, as opposed to RL as a field of study, of which MARL is a subfield.

36th Conference on Neural Information Processing Systems (NeurIPS 2022).

Multi-agent reinforcement learning (MARL) extends RL with capabilities to solve large-scale decentralised decision-making tasks where many agents are expected to coordinate to achieve a shared objective (Foerster, 2018; Oroojlooyjadid and Hajinezhad, 2019; Yang and Wang, 2020). In this cooperative setting, where there is a common goal and rewards are shared between agents, sensible evaluation methodology from the single-agent case often directly translates to the multi-agent case. However, despite MARL being far less developed and entrenched than RL, arguably making adoption of principled evaluation methods easier, the field remains affected by many of the same issues as found in RL. Implementation variance, inconsistent baselines, and insufficient statistical rigour still affect the quality of reported results. Although work specifically addressing these issues in MARL have been rare, there have been recent publications that implicitly observe some of the aforementioned issues and attempt to address their symptoms but arguably not their root cause (Yu et al., 2021; Papoudakis et al., 2021; Hu et al., 2022). These works usually attempt to perform new summaries of performance or look into the code-level optimisations used in the literature to provide insight into the current state of research.

In this paper, we argue that to facilitate long-term progress in future research, MARL could benefit from the literature found in RL on evaluation. The highlighted issues and recommendations from RL may serve as a guide towards diagnosing similar issues in MARL, as well as providing scaffolding around which a standardised protocol for evaluation could be developed. In this spirit, we survey key works in RL over the previous decade, each relating to a particular aspect of evaluation, and use these insights to inform potential standards for evaluation in MARL.

For each aspect of evaluation considered, we provide a detailed assessment of the corresponding situation in MARL through a meta-analysis of prior work. In more detail, our meta-analysis involved manually annotating MARL evaluation methodologies found in research papers published between 2016 to 2022 from various conferences including NeurIPS, ICML, AAMAS and ICLR, with a focus on *deep* cooperative MARL (see Figure 1). In total, we collected data from 75 cooperative MARL papers accepted for publication. Although we do not claim our dataset to comprise the entire field of modern deep MARL, to the best of our knowledge, our data includes all popular and recent deep MARL algorithms and methodologies from seminal papers. We believe this dataset is the first of its kind and we have made it publicly available for further analysis.

By mining the data on MARL evaluation from prior work, we highlight how certain trends, worrying inconsistencies, poor reporting, a lack of uncertainty estimation, and a general absence of proper standards for evaluation, is plaguing the current state of MARL research, making it difficult to draw sound conclusions from comparative studies. We combine these findings with the earlier issues and recommendations highlighted in the literature on RL evaluation, to propose a standardised performance evaluation protocol for cooperative MARL research.

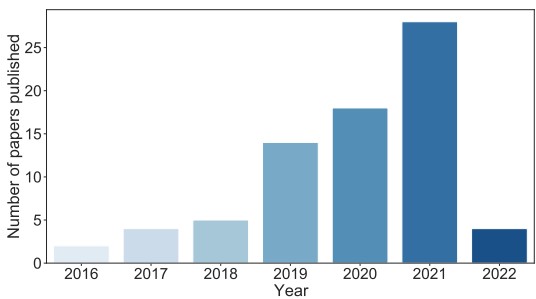

Figure 1: Recorded papers by year in the meta-analysis on evaluation methodologies in cooperative MARL.

In addition to a standardised protocol for evaluation, we expose trends in the use of benchmark environments and suggest useful standards for environment designers that could further improve the state of evaluation in MARL. We touch upon aspects of environment bias, manipulation, level/map cherry picking as well as scalability and computational considerations. Our recommendations pertain to designer specified standards concerning the use of a particular environment, its agreed upon settings, scenarios and version, as well as improved reporting by authors and preferring the use of environments designed to test generalisation.

There are clear parallels between RL and MARL evaluation, especially in the cooperative setting. Therefore, we want to emphasise that our contribution in this work is less focused on innovations in protocol design (of which much can be ported from RL) and more focused on data-driven insights on the current state of MARL research and a proposal of standards for MARL evaluation.

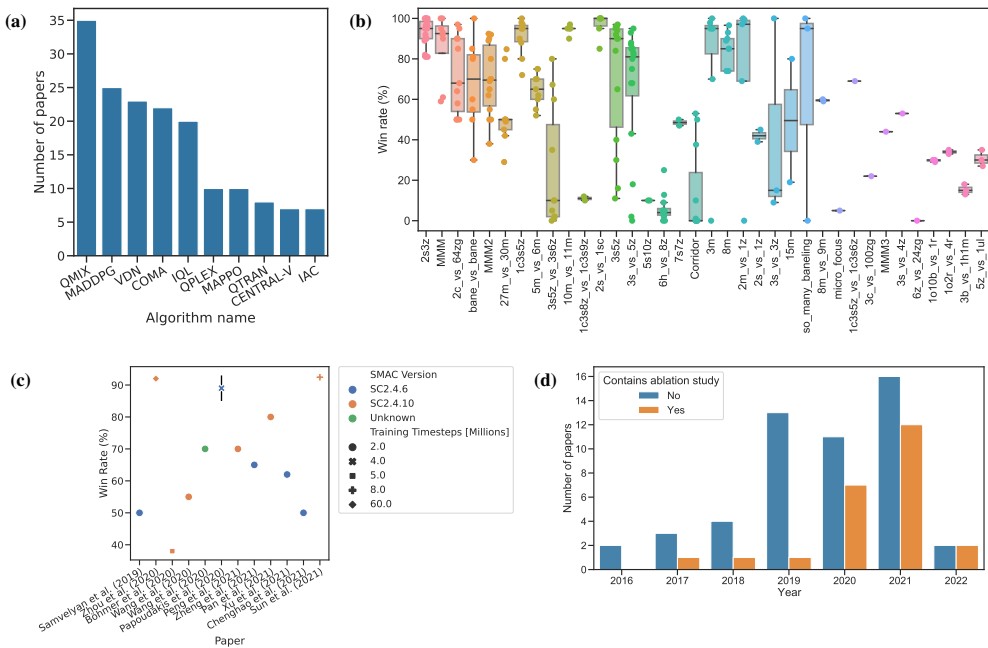

Figure 2: *Inconsistencies in performance reports and a lack of ablation studies*. **(a)** MARL algorithms ranked by popularity. **(b)** Historical performance of QMIX on different SMAC maps across papers. **(c)** Performance of QMIX on the MMM2 SMAC map as reported in different papers. **(d)** A count of papers over the years containing any type of ablation study as part of their evaluation of a newly proposed algorithm.

## 2 From RL to MARL evaluation: lessons, trends and recommendations

In this section, we provide a list of key lessons from RL that are applicable to MARL evaluation. We highlight important issues identified from the literature on RL evaluation, and for each issue, provide an assessment of the corresponding situation and trends in MARL. Finally, we conclude each lesson with recommendations stemming from our analysis and the literature.

### 2.1 Lesson 1: Know the true source of improvement and report everything

**Issue in RL** – *Confounding code-level optimisations and poor reporting*: It has been shown empirically that across some RL papers there is considerable variance in the reported results for the same algorithms evaluated on the same environments (Henderson et al., 2018; Jordan et al., 2020). This variance impedes the development of novel algorithmic developments by creating misleading performance comparisons and making direct comparisons of results across papers difficult. Implementation and code-level optimisation differences can have a significant impact on algorithm performance and may act as confounders during performance evaluation (Engstrom et al., 2020; Andrychowicz et al., 2020). It is rare that these implementation and code-level details are reported, or that appropriate ablations are conducted to pinpoint the true source of performance improvement.

**The situation in MARL** – *Similar inconsistencies and poor reporting but a promising rise in ablation studies*: There already exist some work in MARL showing the effects of specific code-level optimisations on evaluation performance (Hu et al., 2021). Employing different optimisers, exploration schedules, or simply setting the number of rollout processes to be different, can have a significant effect on algorithm performance. To better understand the variance in performance reporting across works in MARL, we focused on QMIX (Rashid et al., 2018), the most popular algorithm in our dataset (as shown in Figure 2 (a)). In Figure 2 (b), we plot the performance of QMIX tested on different maps from the StarCraft multi-agent challenge (SMAC) environment (Samvelyan et al., 2019), a popular benchmark in MARL. On several maps, we find wildly different reported performances with large discrepancies across papers. Although it is difficult to pin down the exact source of these differences in reports, we zoom in with our analysis to only consider a single environment, in this case *MMM2*. We find that some of the variance is explained by differences in the environment version as well as the length of training time, as shown in Figure 2 (c). However, even when both of these aspects are controlled for, as well as any implementation or evaluation

details mentioned in each paper, differences in performance are still observed (as seen by comparing orange and blue circles, respectively). This provides evidence that unreported implementation details, or differences in evaluation methodology account for some of the observed variance and act as confounders when comparing performance across papers (similar inconsistencies in other maps are shown in the Appendix). We finally consider studying the explicit attempts in published works at understanding the sources of algorithmic improvement through the use of ablation studies. We find that very few of these studies were performed in the earlier years of MARL (see Figure 2 (d)). However, even though roughly 40% of papers in 2021 still lacked any form of ablation study, we find a promising trend showing that ablation studies have become significantly more prevalent in recent years.

**Recommendations** – *Report all experimental details, release code and include ablation studies*: Henderson et al. (2018) emphasise that for results to be reproducible, it is important that papers report *all* experimental details. This includes hyperparameters, code-level optimisations, tuning procedures, as well as a precise description of how the evaluation was performed on both the baseline and novel work. It is also important that code be made available to easily replicate findings and stress test claims of general performance improvements. Furthermore, Engstrom et al. (2020) propose that algorithm designers be more rigorous in their analysis of the effects of individual components and how these impact performance through the use of detailed ablation studies. It is important that researchers practice diligence in attributing how the overall performance of algorithms and their underlying algorithmic behavior are affected by different proposed innovations, implementation details and code-level optimisations. In the light of our above analysis, we argue that the situation is no different in MARL, and therefore suggest the field adopt more rigorous reporting and conduct detailed ablation studies of proposed innovations.

## 2.2 Lesson 2: Use standardised statistical tooling for estimating and reporting uncertainty

**Issue in RL** – *Results in papers do not take into account uncertainty*: We have discussed how different implementations of the same algorithm with the same set of hyperparameters can lead to drastically different results. However, even under such a high degree of variability, typical methodologies often ignore the uncertainty in their reporting (Colas et al., 2018a, 2019; Jordan et al., 2020; Agarwal et al., 2022). Furthermore, most published results in RL make use of point estimates like the mean or median performance and do not take into account the statistical uncertainty arising from only using a finite number of testing runs. For instance, Agarwal et al. (2022) found that the current norm of using only a few runs to evaluate the performance of an RL algorithm is insufficient and does not account for the variability of the point estimate used. Furthermore, Agarwal et al. (2022) also revealed that the manner in which point estimates are chosen varies between authors. This inconsistency invalidates direct comparison between results across papers.

**The situation in MARL** – *A lack of shared standards for uncertainty estimation and concerning omissions in reporting*: In Figure 3 (a)-(c), we investigate the use of statistical aggregation and uncertainty quantification methods in MARL. We find considerable variability in the methods used, with little indication of standardisation. Perhaps more concerning is a complete absence of proper uncertainty quantification from one third of published papers. On a more positive note, we observe an upward trend in the use of standard deviation as an uncertainty measure in recent years, particularly in 2021. Furthermore, it has become fairly standard in MARL to evaluate algorithms at regular intervals during the training phase by conducting a certain number of independent evaluation runs in the environment using the current set of learned parameters. This procedure is then followed for each independent training run and results are aggregated to assess final performance. In our analysis, we find that key aspects of this procedure are regularly omitted during reporting, as shown in Figure 3 (d)-(e). Specifically, in (d), we find many papers omit details on the exact evaluation interval used, and in (e), a similar trend in omission regarding the exact number of independent evaluation runs used. Finally, in Figure 3 (f), we plot the number of independent runs used during training, showing no clear standard. Given the often high computational requirements of MARL research, it is not surprising that most works opt for a low number of independent training runs, however this remains of concern when making statistical claims.

**Recommendations** – *Standardised statistical tooling and uncertainty estimation including detailed reporting*: As mentioned before, the computational requirements in MARL research often make it prohibitively difficult to run many independent experiments to properly quantify uncertainty. One approach to make sound statistical analysis more tractable, is to pool results across different tasks

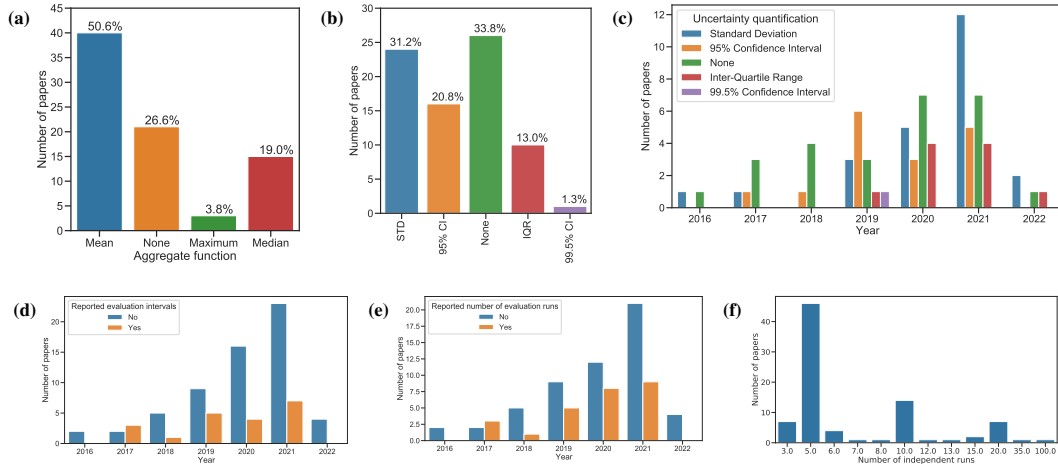

Figure 3: *Trends in performance aggregation, uncertainty quantification and omissions in reporting on key aspects of evaluation methodology*. **(a)** Distribution of performance aggregation metrics. None means that the aggregation metric was not specified in the paper. **(b)** Distribution of uncertainty methods: $\alpha\%$ confidence interval (CI), standard deviation (STD) and interquartile range (IQR). **(c)** Trends in uncertainty quantification over time. **(d)** Reporting of the evaluation interval used. **(e)** Reporting on the number of evaluation runs used. **(f)** Number of independent runs used in published work.

using the bootstrap (Efron, 1992). In particular, for RL, Agarwal et al. (2022) recommend computing stratified bootstrap confidence intervals, where instead of only using the original set of data to calculate confidence intervals, the data is resampled with replacement from $M$ tasks, each having $N$ runs. This process is repeated as many times as needed to approximate the sampling distribution of the statistic being calculated. Furthermore, when making a summary of overall performance across tasks it has been shown that the mean and median are insufficient, the former being dominated by outliers and the latter having higher variance. Instead, Agarwal et al. (2022) propose the use of the interquartile mean (IQM) which is more robust to outlier scores than the mean and more statistically efficient than the median. Finally, Agarwal et al. (2022) propose the use of probability of improvement scores, sample efficiency curves and performance profiles, which are commonly used to compare the performance of optimization algorithms (Dolan and Moré, 2001). These performance profiles are inherently robust to outliers on both ends of the distribution tails and allow for the comparison of relative performance at a glance. In the shared reward setting, where it is only required to track a single return value, we argue that these tools fit the exact needs of cooperative MARL, as they do in RL. Furthermore, in light of our analysis, we strongly recommend a universal standard in the use, and reporting of, evaluation parameters such as the number of independent runs, evaluation frequency, performance metrics and statistical tooling, to make comparisons across different works easier and more fair.

### 2.3   Lesson 3: Guard against environment misuse and overfitting

**Issue in RL** – *Over-tuning algorithms to perform well on a specific environment*: As early as 2011 issues with evaluation in RL came to the foreground in the form of *environment overfitting*. Whiteson et al. (2011) raised the concern that in the context of RL, researchers might over-tune algorithms to perform well on a specific benchmark at the cost of all other potential use cases. More specifically, Whiteson et al. (2011) define environment overfitting in terms of a desired target distribution. When an algorithm performs well in a specific environment but lacks performance over a target *distribution* of environments, it can be deemed to have overfit that particular environment.

**The situation in MARL** – *One environment to rule them all – on the use and misuse of SMAC*: The StarCraft multi-agent challenge (SMAC) (Samvelyan et al., 2019) has quickly risen to prominence as a key benchmark for MARL research since it's release in 2019, rivaled only in use by the multi-agent particle environment (MPE) introduced by Lowe et al. (2017) (see Figure 4 (a)). SMAC and its accompanied MARL framework PyMARL (introduced in the same paper), fulfills several desirable properties for benchmarking: offering multiple maps of various difficulty that test key aspects of MARL algorithms and providing a unified API for running baselines as well as state-of-the-art algorithms on SMAC. Unfortunately, the wide-spread adoption of SMAC has also caused several issues, relating to environment overfitting and cherry picking of results, putting into question the

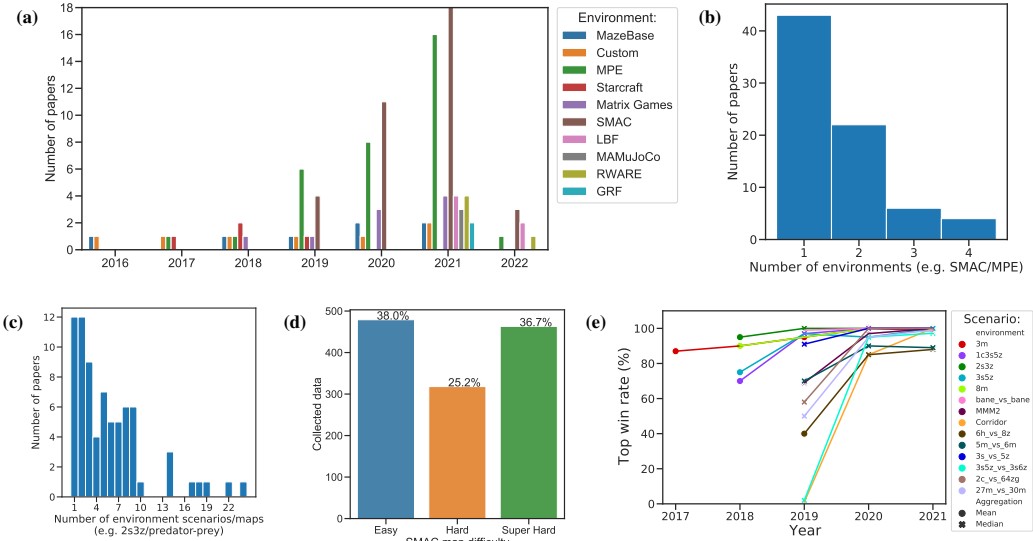

Figure 4: *Environment popularity, usage trends in papers and potential evidence of overfitting on SMAC.* **(a)** Environment adoption over time. **(b)** Number of environments used in papers. **(c)** Number of scenarios/tasks/maps used in papers. **(d)** Distribution of task difficulty of SMAC maps used in papers. **(e)** Performance trends on popular SMAC maps: Aggregation is the aggregate function used for the different reported values.

credibility of claims made while using it as a benchmark.

To illustrate the above point, we start by highlighting that many MARL papers use only a single environment (e.g. SMAC or MPE) for evaluation, as shown in Figure 4 (b). This is often deemed acceptable since both SMAC and MPE provide many different tasks, or maps. For instance, in SMAC, there are 23 different maps providing a wide variety in terms of the number of agents, agent types and game dynamics. However, there is no standard, or agreed upon set of maps to use for benchmarking novel work, which makes it easy for authors to selectively subsample maps post-experiment based on the outcomes of their proposed algorithm. As shown in Figure 4 (c), although environments like SMAC and MPE offer many different testing scenarios, it is typical for papers to only use a small number of these in their reported experiments.

To concretely expose the potential danger in this author map selection bias, we redo the original analysis performed by Samvelyan et al. (2019), using the authors' exact experimental data that was made publicly available[6], containing five independent runs for IQL (Tampuu et al., 2015), COMA (Foerster et al., 2018), VDN (Sunehag et al., 2017) and QMIX (Rashid et al., 2018) on 14 SMAC maps.[7] The top row of Figure 5 shows the results of this analysis performed using the statistical tools recommended by Agarwal et al. (2022), including the probability of improvement between algorithms, performance profiles and sample efficiency curves. The results support the original claims made by Samvelyan et al. (2019), namely that QMIX is a superior algorithm to that of VDN, COMA and IQL, both in terms of performance and sample efficiency. However, by simply sampling a smaller set of two easy, medium and hard maps (a common spread in the literature, see Figure 4 (d)), from the original 14, giving 6 maps in total (a reasonable number according to prior work, see Figure 4 (c)), we are able to change the outcome of the analysis in support of no difference in performance between VDN and QMIX, as well as finding VDN to be more sample efficient. This is shown in the bottom row of Figure 5 and highlights the danger of a lack of standards regarding which *fixed* set of maps should be used for benchmarking.

We end our investigation into SMAC (and refer the reader to the Appendix for additional discrepancies uncovered), by looking at historical performance trends. In Figure 4 (e), we show the top win rate achieved by an algorithm in a specific year for 14 of the most popular maps used in prior work. We find that by 2021, most of these maps have converged to a win rate close or equal to 100%, while only a few maps are still situated around 80-90%. Given that many of these maps repeatedly feature

[6]We applaud the authors for making their raw evaluation data publicly available. This data can be found here: https://github.com/oxwhirl/smac.

[7]The maps chosen were: 1c3s5z, 2c_vs_64zg, bane_vs_bane, MMM2, 10m_vs_11m, 27m_vs_30m, 5m_vs_6m, 2s3z, 2s_vs_1sc, s5z, 3s_vs_5z, 6h_vs_8z, 3s5z_vs_3s6z and corridor

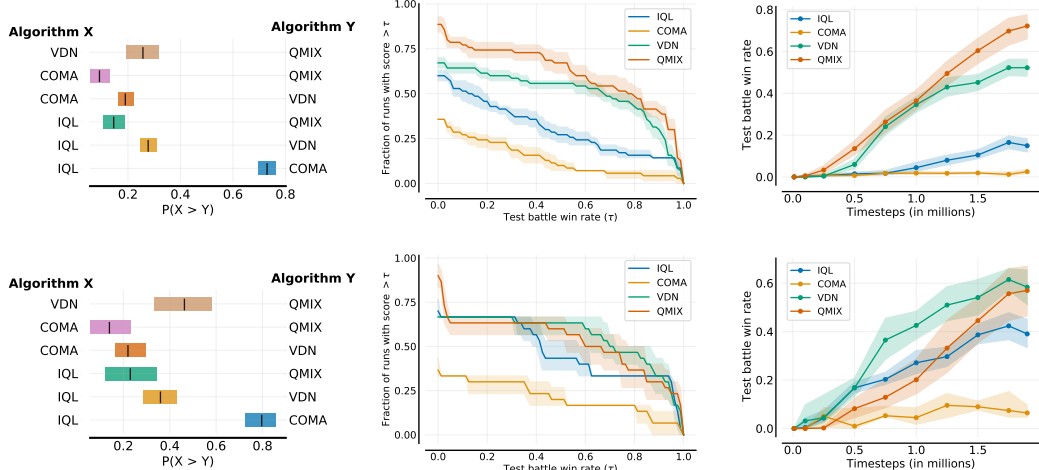

Figure 5: *Reanalysis of original SMAC experiments conducted by Samvelyan et al. (2019) including probability of improvement, performance profiles and sample efficiency curves.* **Top row:** All 14 SMAC maps used in the original analysis. **Bottom row:** Subset of 6 maps, including 2 easy, 2 medium and 2 hard: 2s_vs_1sc, 3s_vs_5z, bane_vs_bane, 5m_vs_6m, 6h_vs_8z and corridor

across papers, and will likely be used in future work, it begs the question to what extent the MARL community has already overfit to SMAC as an evaluation benchmark.

**Recommendations** – *Standardised environment sets and testing for generalisation*: To solve the issue of environment overfitting, Whiteson et al. (2011) propose the use of a generalised evaluation methodology. In this approach, environments (for tuning algorithms) are freely sampled from some generalised environment set. Separately, algorithm evaluation is performed on a second set of sampled environments from the same generalised environment set, acting as a test set analogous to that used in supervised learning. Recent work in this direction include benchmarks such as Procgen (Cobbe et al., 2020), which use procedural generation to implicitly construct a distribution from which to sample test tasks. In SMAC and other environments, it is common practice to only evaluate on the exact map the algorithm was trained on, under that exact same conditions, and to not specifically test for generalisation across unseen tasks. However, many MARL algorithms are still highly sensitive to small changes in the environment and often fail to generalise to new unseen tasks (Carion et al., 2019; Zhang et al., 2020; Mahajan et al., 2022). This calls for more work on MARL generalisation and we recommend a stronger focus on benchmarks designed specifically to test generaliation. However, it has been noted that procedurally generated benchmarks may reduce the precision of research (Kirk et al., 2021), making it more difficult to track progress. Furthermore, MARL exibits several unique and challenging difficulties when it comes to building algorithms able to generalise, likely requiring many years of future work to surmount (Mahajan et al., 2022). Therefore, in certain cases, it might make more sense for researchers to take smaller and more precise steps towards key innovations in algorithm design by still relying on traditional environment sets. In this setting, we strongly advocate using fixed environment sets, where ideally these are selected by the designers of each environment and are accompanied by exact instruction for their configuration, so as to be consistent across papers.

## 3 Towards a standardised evaluation protocol for MARL

In this section, we pool together the observations and recommendations from the previous section to provide a standardised performance evaluation protocol for cooperative MARL. We are realistic in our efforts, knowing that a single protocol is unlikely to be applicable to all MARL research. However, echoing recent work on evaluation (Ulmer et al., 2022), we stress that many of the issues highlighted previously stem from a lack of *standardisation*. Therefore, we believe a default "off-the-shelf" protocol that is able capture most settings, could provide great value to the community. If widely adopted, such a standardised protocol would make it easier and more accurate to compare across different works and remove some of the noise in the signal regarding the true rate of progress in MARL research. A summarised version of our protocol is given in the blue box at the end of this section and a concrete demonstration of its usage can be found in the Appendix.

**Benchmarks and Baselines.** Before giving details on the proposed protocol, we first briefly comment on benchmarks and baselines used in experiments. These choices often depend on the research question of interest, the novel work being proposed and key algorithmic capabilities to be tested. However, as alluded to in our analysis, we recommend that environment designers take full ownership regarding how their environments are to be used for evaluation. For example, if an environment has several available static tasks, the designers should specify a fixed compulsory minimum set for experiments to avoid biased subsampling by authors. It could also be helpful if designers keep track of state-of-the-art (SOTA) performances on tasks from published works and allow authors to submit these for vetting. We also strongly recommend using more than a single environment (e.g. SMAC) and preferring environments that test generalisation. Regarding baselines, we recommend that at minimum the published SOTA contender to novel work should be included. For example, if the novel proposal is a value-based off-policy algorithm for discrete environments, at minimum, it must be compared to the current SOTA value-based off-policy algorithm for discrete environments. Finally, all baselines must be tuned fairly with the same compute budget.

**Evaluation parameters.** In Figure 6, we show the evaluation parameters for the number of evaluation runs, the evaluation interval (top) and the training time (bottom left) used in papers. We find it is most common to use 32 evaluation episodes at every 10000 timesteps (defined as the steps taken by agents when acting in the environment) and to train for 2 million timesteps in total. We note that these numbers are skewed towards earlier years of SMAC evaluation and that recent works have since explored far longer training times. However, we argue that these longer training times are not always justified (e.g. see the bottom right of Figure 6). Furthermore, SMAC is one of the most expensive MARL environments (Appendix A.7 in Papoudakis et al. (2021)) and for the future accessibility of research in terms of scale and for fair comparisons across different works, we recommend the above commonly used values as reasonable starting defaults and support the

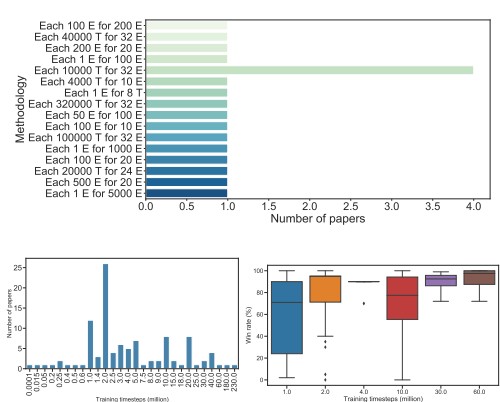

Figure 6: **Top:** Evaluation methodologies in papers: how frequent and for how many runs to do evaluation. "E" refers to episode and "T" to timestep. **Bottom:** Number of total training timesteps used in papers (left) and win rates per training timesteps on 2s3z (right).

view put forward by Dodge et al. (2019) that results should be interpreted as a function of the compute budget used. We of course recognise that these evaluation parameters can be very specific to the environment, or task, and again we urge environment designers to play a role in helping the community develop and adopt sensible standards.

**Performance and uncertainty quantification.** To aggregate performance across evaluation intervals, we recommend using the *absolute* performance metric proposed by Colas et al. (2018b), computed using the best average (over training runs) joint policy found during training. Typically, practitioners checkpoint the best performing policy parameters to use as the final model, therefore it makes sense to do evaluation in a similar way. However, to account for not averaging across different evaluation intervals, Colas et al. (2018b) recommend increasing the number of independent evaluation runs using the best policy by a factor of 10 compared to what was used at other intervals. To quantify uncertainty, we recommend using the mean with 95% confidence intervals (CIs) at each evaluation interval (computed over independent evaluation runs), and when aggregating across tasks within an environment, we recommend using the tools proposed by Agarwal et al. (2022), in particular, the inter-quartile mean (IQM) with 95% stratified Bootstrap CIs.

**Reporting.** We strongly recommend reporting *all* relevant experimental details including: hyperparameters, code-level optimisations, computational requirements and frameworks used. Taking inspiration from *model cards* (Mitchell et al., 2019), we provide templates for reporting in the Appendix. Furthermore, we recommend providing experimental results in multiple formats, including plots *and* tables per task and environment as well as making all raw experimental data and code publicly available for future analysis and easy comparison. Finally, we encourage authors to include detailed ablation studies in their work, so as to be able to accurately attribute sources of improvement.

<div class="box">

**A Standardised Performance Evaluation Protocol for Cooperative MARL**

**Input:** Environments with tasks $t$ from a set $\mathcal{T}$. Algorithms $a \in \mathcal{A}$, including baselines and novel work.

**1. Evaluation parameters – defaults**

- Number of training *timesteps*, $T = 2$ million.
- Number of independent training *runs*, $R = 10$ (from Agarwal et al. (2022))
- Number of independent evaluation *episodes* per interval, $E = 32$.
- Evaluation *intervals*, $i \in \mathcal{I}$, at every 10000 timesteps.

**2. Performance and uncertainty quantification**

1. Performance metric: Always use returns $G$ (applicable to all environments), *and* the environment specific metric (e.g. Win rate).

2. Per task evaluation: Compute the mean $G_t^a$ over $E$ episodes at each evaluation interval $i$, where $G_t^a$ is the return of algorithm $a$ on task $t$, with 95% CI, for all $a$.

3. Per environment evaluation:

   - Compute the normalised *absolute* return (Colas et al., 2018b) as the mean return of $10 \times E = 320$ evaluation episodes using the best joint policy found during training and normalising the return to be in the range $[0, 1]$ using $(G_t^a - \min(G_t))/(\max(G_t) - \min(G_t))$, where $G_t$ is the return for all algorithms on task $t$.
   - For each algorithm $a$, form an evaluation matrix with shape $(R, |\mathcal{T}|)$ where each entry is the normalised absolute return for a specific training run on a specific task.
   - Compute the IQM and optimality gap with 95% stratified Bootstrap CIs, probability of improvement scores and performance profiles, to compare the algorithms, using the tools proposed by Agarwal et al. (2022).[a] Sample efficiency curves can be computed by using normalised returns at each evaluation interval.

**3. Reporting**

- Experiments: All hyperparameters, code-level optimisations, computational requirements and framework details.
- Plots: All task and environment evaluations as well as ablation study results.
- Tables: Normalised absolute performance per task with 95% CI for all tasks, IQM with 95% stratified Bootstrap CIs per environment for all environments.
- Public repository: Raw evaluation data and code implementations.

---

[a]These can be found in the `rliable` library: https://github.com/google-research/rliable

</div>

# 4   Conclusion and future work

In this work, we argue for the power of standardisation. In a fast-growing field such as MARL, it becomes ever more important to be able to dispel illusions of rapid progress, potentially misleading the field and resulting in wasted time and effort. We hope to break the spell by proposing a sensible standardised performance evaluation protocol, motivated in part by the literature on evaluation in RL, as well as by a meta-analysis of prior work in MARL. If widely adopted, such a protocol could make comparisons across different works much faster, easier and more accurate. However, certain aspects of evaluation are better left standardised outside of the control or influence of authors, such as protocols pertaining to the use of benchmark environments. We believe this is an overlooked issue and an important area for future work by the community, and specifically environment designers, to jointly establish better standards and protocols for evaluation and environment use. Finally, we encourage the community to move beyond the use of only one or two environments with static task sets (e.g. MPE and SMAC) and focus more on building algorithms, environments and tools for improving generalisation in MARL.

A clear limitation of our work is our focus on the cooperative setting. Interesting works have developed protocols and environments for evaluation in both the competitive and mixed settings (Omidshafiei et al., 2019; Rowland et al., 2019; Leibo et al., 2021). We find this encouraging and argue for similar efforts in the adoption of proposed standards for evaluation.

## Acknowledgments and Disclosure of Funding

The authors would like to kindly thank the following people for useful discussions and feedback on this work: Jonathan Shock, Matthew Morris, Claude Formanek, Asad Jeewa, Kale-ab Tessera, Sinda Ben Salem, Khalil Gorsan Mestiri, Chaima Wichka and Sasha Abramowitz.

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
