# OpenReview forum: "Towards a Standardised Performance Evaluation Protocol for Cooperative MARL"
_NeurIPS.cc/2022/Conference — NeurIPS 2022 Accept_

### Official Review · Reviewer_F87n · 2022-07-02

**Rating:** 6
**Confidence:** 3
**Soundness:** 4 excellent
**Presentation:** 3 good
**Contribution:** 3 good

**Summary:**


Towards a Standardized Performance Evaluation Protocol for Cooperative MARL
In this paper, the authors review existing papers on multi-agent reinforcement learning, and show three lessons from these papers: 1) we need to study what exactly makes the improvement and don’t hide details; 2) we need to have standardized results reporting and uncertainty analysis; 3) we need to avoid exploiting the environments and overfitting it.
And authors also propose a standard evaluation protocol for future MARL research.


**Questions:**

1) What does the author believe are the problems in multiagent RL that are different or unique as opposed to the ones also commonly seen in general RL research?
2) What do the authors propose to help all the researchers to follow the correct protocol and practices in RL? Is there a code-base or website MARL researchers could use for MA benchmarks?
3) There’s a very related paper here [4] (Revisiting Some Common Practices in Cooperative Multi-Agent Reinforcement Learning). This is a paper that was available very recently, but the author might want to cite it as a contemporary work in the final version.




**Limitations:**


I don’t find anything that would potentially have a negative societal impact.

[1] Agarwal, Rishabh, Max Schwarzer, Pablo Samuel Castro, Aaron C. Courville, and Marc Bellemare. "Deep reinforcement learning at the edge of the statistical precipice." Advances in neural information processing systems 34 (2021): 29304-29320.
[2] Henderson, Peter, Riashat Islam, Philip Bachman, Joelle Pineau, Doina Precup, and David Meger. "Deep reinforcement learning that matters." In Proceedings of the AAAI conference on artificial intelligence, vol. 32, no. 1. 2018.
[3] Whiteson, Shimon, Brian Tanner, Matthew E. Taylor, and Peter Stone. "Protecting against evaluation overfitting in empirical reinforcement learning." In 2011 IEEE symposium on adaptive dynamic programming and reinforcement learning (ADPRL), pp. 120-127. IEEE, 2011.
[4] Fu, Wei, Chao Yu, Zelai Xu, Jiaqi Yang, and Yi Wu. "Revisiting Some Common Practices in Cooperative Multi-Agent Reinforcement Learning." arXiv preprint arXiv:2206.07505 (2022).



**Strengths And Weaknesses:**


Originality: The paper discusses a very important topic in multi-agent RL that draws more and more attention recently. Some of the findings, for example the inconsistent reported performance, hidden performance tricks under the sleeves, improper handling of results reporting and missing uncertainty evaluations, have been already discovered as problems in general RL research. That being said, a lot of the findings have already been discovered in Agarwal et al. (2022), Whiteson et al. (2011), Henderson et al. (2018).

On the bright side, the paper gives a very good systematic review on multi-agent reinforcement learning algorithms. Figures such as Figure 2, Figure 4 are quite inspiring and the discussion about QMIX, VDN, COMA and IQL give quite valuable information about the actual performance of these algorithms.
The paper also provides a concluding protocol suggestion, which will benefit the community.


Quality and clarity: The paper is well written, supported with informative figures and plots.
Each section has a very clear topic and the authors provide detailed explanations for every proposal in the paper.

Significance: A lot of the problems have been discussed by other related papers, which potentially will make the impact of this paper less. However the proposed protocol is beneficial to the multi-agent RL community.

---

> ### Author Response · Authors · 2022-07-29
> **Reply to reviewer F87n**
>
> We kindly thank the reviewer for their feedback as well as their insightful questions and suggestions. We would like to take this opportunity to address each of them in turn.
>
> * “What does the author believe are the problems in multiagent RL that are different or unique as opposed to the ones also commonly seen in general RL research?”
>
> In general, MARL poses many problems unique to its problem setting that are not found in single-agent RL research. Particularly from the perspective of algorithm design and methods that are capable of dealing with challenges specific to the multi-agent case, including non-stationarity (in the environment from the perspective of each agent, since other agents are also learning), credit assignment (between agents at a particular time step as well as temporally across timesteps), partial observability (in decentralised control settings) and the need for complex coordination. MARL also has a unique set of application domains and research benchmarks.  However, although in the cooperative setting these challenges and domains remain unique to MARL, evaluation is more similar to single-agent RL. But as we note in our general reply (which we encourage the reviewer to please also read) before this work, it was not clear to what extent the current state of evaluation in MARL was plagued by similar issues. Having now knowledge of this, it highlighted the need and motivation for our proposed standardised evaluation protocol.
>
> * “What do the authors propose to help all the researchers to follow the correct protocol and practices in RL? Is there a code-base or website MARL researchers could use for MA benchmarks?”
>
> We want to specifically thank the reviewer for asking this question. We have indeed developed specific resources to help researchers and have listed them in our general reply (please see above), all of which will be publicly released in full, upon publication. That said, we envision our efforts on this research topic to continue into future work that will see more guidelines and tools being developed to assist the community even further and we will welcome contributions from the community as well.
>
> * “There’s a very related paper here [4] (Revisiting Some Common Practices in Cooperative Multi-Agent Reinforcement Learning). This is a paper that was available very recently, but the author might want to cite it as a contemporary work in the final version.”
>
> We thank the reviewer for pointing us to this paper.

---

### Official Review · Reviewer_8mDm · 2022-07-07

**Rating:** 8
**Confidence:** 4
**Soundness:** 4 excellent
**Presentation:** 4 excellent
**Contribution:** 4 excellent

**Summary:**

This paper reveals the trends and worrying issues of evaluation via an extensive study of recent MARL works. Several concerns are raised regarding inconsistent results, poor statistical aggregation, and environmental misuse.  Based on the analysis, the authors propose several recommendations and a standardized protocol, specifying the methodologies of how evaluation parameters selection, uncertainty estimation, and reporting should be performed.

**Questions:**

+ In L280, by using the term "timesteps", do the authors mean environment frames or training steps? What is the batch size and the frequency of updating parameters?
+ L191: "SMAC and it's" -> "SMAC and its"
+ L289: please use \citet

**Limitations:**

I don't know much about the works investigating the evaluation protocol in single-agent RL. It seems that the authors just translate the designed protocols into the domain of cooperative MARL. However, as the authors mentioned in the introduction section, the contribution of this paper is not an innovation of evaluation protocol. This limitation can be minor.

**Strengths And Weaknesses:**

## Strengths

+ The logic of this paper is easy to follow, and the meta-analysis is comprehensive.

+ The summary and proposal of a standardized evaluation protocol is valuable.

As a practitioner, I quite understand that reproducibility is a huge problem in recent MARL works. It is worthwhile (however difficult) to summarize the evaluation methods applied in these works. I really appreciate the efforts of the authors and hope the standardized protocol could be utilized by researchers in the field.

## Weaknesses

I don't find any significant weaknesses in this paper.

---

> ### Author Response · Authors · 2022-07-29
> **Reply to reviewer 8mDm**
>
> We kindly thank the reviewer for their positive feedback and suggested corrections, all of which have been implemented in the revised version of the paper. We appreciate the reviewer’s views on the importance of this work, especially from the perspective of a MARL practitioner, which resonates closely with our own experience. It is also our hope that the proposed standardised protocol will be adopted by researchers in the field, making it easier for the community to accurately measure its progress. Finally, we invite the reviewer to please read our message regarding the resources we will be releasing to support the community and for more information on how to apply this protocol in their own work.

---

### Official Review · Reviewer_qdGs · 2022-07-08

**Rating:** 7
**Confidence:** 3
**Soundness:** 3 good
**Presentation:** 4 excellent
**Contribution:** 3 good

**Summary:**

This paper is a interesting meta-research of cooperative MARL methods. The authors claim at least three major problems in current MARL research: report of implementation and evaluation details, report of uncertainty with proper statistical tool, the choice of benchmark environments. The paper tries to appeal for a standard protocol for MARL experiments, which should be encouraged for benefiting the community.

**Questions:**

For ablation study analysis in Sec. 2.1, it seems only the number of papers using ablation study are discussed. Is there any deeper investigation of the number and types of ablation study within each paper, and what should be a standard for ablation study?

The results in Fig.4 (e) show that some methods even have a decreasing performance along the time. Does that mean the later work does not report the performance as good as the original paper claims?

**Limitations:**

As mentioned in the weakness, more insights of problems specific for MARL settings are interesting to know, apart from the commonly known single-agent RL problems.

Although the authors propose a performance evaluation protocol for cooperative MARL in Sec. 3, some other problems raised by authors are not concretely solved, like how to choose a standard set of environments for different MARL settings, etc. I understand this problem may be hard to give an answer and the authors may want to receive more feedback from the community. I think if this paper is the one trying to establish the protocol for MARL experiments, these problems are better solved with concrete solutions even if in the follow-up works.

**Strengths And Weaknesses:**

This paper conduct heavy statistical analysis of previous MARL algorithms, which are reported in the diagrams. The results are clear and thorough. The efforts of the authors should be appreciated. The description of the problems are clear. More importantly, recommendations for solving the problems are provided after each section.

One major weak side of the work is that it seems all three aspects of lessons are inherited from single-agent RL, with no additional components specifically caused by cooperative MARL settings. The problems of reporting implementation details, statistical uncertainty estimation and choice of environments are existing problems in single-agent RL, which makes this work looks like a replicate of commonly known problems in single-agent RL to MARL domain. It could be more interesting to see at least one or two more lessons specifically considered in MARL setting.

---

> ### Author Response · Authors · 2022-07-29
> **Reply to reviewer qdGs**
>
> We kindly thank the reviewer for their encouraging feedback, their insightful comments and questions. We would like to take this opportunity to address each of them in turn.
>
> * “One major weak side of the work is that it seems all three aspects of lessons are inherited from single-agent RL”
>
> Although we address this concern in more detail in our general reply (and we please ask the reviewer to read our full reply there), we provide a quick summary here. In short, although we agree that all three lessons are inherited from single-agent RL, our main contributions are: (1) exposing the extent and severity of these issues in MARL through our meta-analysis and showing the primary cause to be a lack of standards and (2) proposing a solution in the form of a standardised protocol. We therefore feel that this mapping from the single-agent to the multi-agent case does not detract from our contributions, specifically for evaluation, where the two settings are often very similar.
>
> * “The results in Fig.4 (e) show that some methods even have a decreasing performance along the time. Does that mean the later work does not report the performance as good as the original paper claims?”
>
> This is indeed what we have observed in the data. For instance, differences in reported performance of the same baselines between papers. A possible explanation for this could be that when a new algorithm becomes a baseline in a later paper the new authors do not optimise it as well. Another explanation could be differences in code level optimisations between the versions of the same algorithm used in different papers as a baseline, causing differences in the reported results. In fact, there can be many reasons and without standards it is hard to extract the signal from the noise, which is one of the main messages of our paper. One specific aspect of the plot we can concretely comment on is that it includes all data points we collected on these environments as reported by the original work. This means that in some cases, the lack of observed monotonicity can be explained by differences in the evaluation protocol between papers. For example, some papers reporting best performance, versus mean or median performance. However, even when controlling for these differences in methodology, the general trend towards overfitting and in some cases the observed non-monotonicity across time, still remains.
>
> * “Is there any deeper investigation of the number and types of ablation study within each paper, and what should be a standard for ablation study?”
>
> In our analysis, we did not record the number or type of ablations since these seemed to be very specific to each paper containing some form of ablation. However, we were able to identify broadly 5 different types of ablations typically performed in papers:
> 1. Authors introduce multiple improvements which are not reliant on each other and test their effectiveness independently on existing baselines.
> 2. Authors propose a new method that has multiple hyperparameters that have different effects and perform a sweep to determine how these interact with performance.
> 3. Authors propose a new method that has multiple components which can be selectively activated and the performance reduction from removing each component is tested.
> 4. Authors propose a new method in some system that runs on top of existing algorithms and test performance changes from the method’s inclusion on different methods.
> 5. Authors propose a method that has some potential performance modifier like parameter space size and they “upgrade” the baselines to accommodate for this.
>
> Future work, could potentially use these broad categories to update the dataset with information on ablations and extend the protocol with concrete steps for performing meaningful ablations based on the category in which the work falls.
>
> * “some other problems raised by authors are not concretely solved, like how to choose a standard set of environments for different MARL settings. …  I think if this paper is the one trying to establish the protocol for MARL experiments, these problems are better solved with concrete solutions even if in the follow-up works.”
>
> This is certainly true, there remains important problems still unsolved. However, we feel that at least through the efforts of analysis we were able to expose many of these important issues in our paper. A clear first step. Now that the community can be made aware of them, future work can focus on this (previously neglected) area of evaluation, i.e. environment standardisation and usage protocols. The issues here are many, and require a coordinated effort from the community to solve all of them since many of these benchmark environments are developed in a distributed manner across different members of the community. We are excited to continue this line of work in collaboration with the rest of the community to help find meaningful and effective solutions to these remaining problems.

---

> > ### Comment · Reviewer_qdGs · 2022-08-07
> > **Response to Author Rebuttal**
> >
> > I would like to thank the authors for their detailed reply.
> >
> > "For example, some papers reporting best performance, versus mean or median performance." If this is the case, the values in the plot are better associated with the metrics, otherwise it could be an unfair comparison for different reported values.
> >
> > I've read the reply on other problems I raised, and I would encourage the authors to further improve the overall protocols on those directions.

---

> > > ### Author Response · Authors · 2022-08-08
> > > **Response to Reviewer qdGs**
> > >
> > > Firstly, we would like to express our gratitude to the reviewer for taking the time to go through our reply.
> > >
> > > “If this is the case, the values in the plot are better associated with the metrics, otherwise it could be an unfair comparison for different reported values.”
> > >
> > > We appreciate the reviewer's comments towards fairness when comparing diverse works. Taking this feedback into consideration, we updated Fig 4.e to include information about the metric that was used for each data point. Furthermore, we removed two data points (1) the value for 3m in 2017 (due to the metric being the maximum win rate obtained in the environment by the paper in question) and (2) the value for the corridor map in 2019 (due to no aggregation detail being available).  We have included the updated plot in our revised submission and although we have made these refinements to the plot, our original analysis that algorithms are overfitting to the SMAC benchmark remains unchanged.
> > >
> > > “I've read the reply on other problems I raised, and I would encourage the authors to further improve the overall protocols on those directions.”
> > >
> > > In this regard, we agree with the reviewer and look toward the upgrades and refinements that the protocol will undergo over time as a result of feedback from the MARL community. We believe that the dataset, website, tools, and feedback form will serve as a means to streamline such collaborative efforts.

---

### Official Review · Reviewer_hmEq · 2022-07-09

**Rating:** 5
**Confidence:** 3
**Soundness:** 3 good
**Presentation:** 4 excellent
**Contribution:** 2 fair

**Summary:**

This paper's main contribution is the proposed standardized performance evaluation protocol for cooperative MARL. Specifically, this paper performs meta-analyses and identifies the following potential issues in related works:
1. Lack of experimental details, open-sourced code, and ablation studies (Section 2.1)
2. Lack of standardized statistical tools and uncertainty estimation (Section 2.2)
3. Lack of standardized environment sets for preventing domain overfitting (Section 2.3)

Based on these observations, this paper provides the standardized evaluation protocol for cooperative settings in Section 3, including benchmark/baseline, evaluation parameters, statistical quantification, and reporting.

**Questions:**

1. (Related to Weakness) I understand that extending to general-sum/competitive settings is a future work, but I hope to ask for a clarification on why this paper performed meta-analyses limited to cooperative settings. In particular, the main differences between RL and MARL result from the fact that there are multiple learning agents with non-stationary policies. Hence, establishing the standard protocol for general-sum and/or competitive games is an open question because it is often not clear when the learning should end under non-stationarity compared to cooperative settings. As noted in the paper, there are close similarities between RL and cooperative MARL evaluation, and the related works of Henderson (2018) and Agarwal (2021) focus on single-agent settings. As such, if this paper could provide a general protocol that includes general-sum and/or competitive settings, the contribution could have been more significant.
2. In Figure 3(f), there might be a typo in the x-axis label: "Year" to "Number of independent runs"

**Strengths And Weaknesses:**

**Strengths:**
1. The performed meta-analyses provide interesting perspectives on recent trends in cooperative MARL, such as types of MARL algorithms used in papers, ablation studies, and types of performed uncertainty quantification.
2. In general, I agree with the main points of this paper that MARL may lack standardization and thus establishing the standard protocol is important.
3. The paper is written well and conveys the main points clearly.

**Weakness:**
While I agree with the main points, my main concern is the novelty of this paper compared to related works. In particular, the importance of reporting experimental details (Section 2.1) and computing uncertainty estimation (Section 2.2) is already discussed in prior works (Henderson, 2018; Agarwal, 2021), so these aspects have been considered in recent MARL papers. Compared to these works, the domain overfitting issue discussed in Section 2.3 adds a new perspective to the MARL community, but I am unsure whether this perspective alone has sufficient novelty for acceptance.

**References:**
Henderson, et al., Deep Reinforcement Learning that Matters, AAAI 2018
Agarwal, et al., Deep Reinforcement Learning at the Edge of the Statistical Precipice, NeurIPS 2021

---

> ### Author Response · Authors · 2022-07-29
> **Reply to reviewer hmEq**
>
> We kindly thank the reviewer for their constructive feedback as well as their insightful comments and questions. We would like to take this opportunity to address each of them in turn.
>
> * “my main concern is the novelty of this paper compared to related works. In particular, the importance of reporting experimental details (Section 2.1) and computing uncertainty estimation (Section 2.2) is already discussed in prior works (Henderson, 2018; Agarwal, 2021), so these aspects have been considered in recent MARL papers”
>
> We agree with the reviewer that these important topics have already been explored in prior works, but only in the context of single-agent RL and not in MARL. Of course, we acknowledge the parallels between the single agent case and the cooperative MARL setting, but we would still want to emphasize that prior to our work, it was not clear to what extent similar issues (as identified in the above mentioned prior works from single agent RL), were present in MARL. Furthermore, as mentioned in our general reply, core to our contribution is bringing to the attention of the community the severity of these issues in MARL. Many of which are significant and ultimately stem from a lack of standardisation, to which our proposed protocol provides a potential solution.
>
> * “I understand that extending to general-sum/competitive settings is a future work, but I hope to ask for a clarification on why this paper performed meta-analyses limited to cooperative settings. In particular, the main differences between RL and MARL result from the fact that there are multiple learning agents with non-stationary policies. Hence, establishing the standard protocol for general-sum and/or competitive games is an open question because it is often not clear when the learning should end under non-stationarity compared to cooperative settings.”
>
> Our full data collection included all settings: cooperative, competitive and general-sum. However, after collection, we found that more than 90% of papers that we collected were in the cooperative setting. Therefore, cooperative MARL represented by far the most popular setting and captured most of the current research efforts in the field. Because of this, and our motivation to construct a standardised protocol based on statistical trends, we opted to only focus on the cooperative setting for this work. However, we still acknowledge the importance of other settings and feel this is an important area of future work, not only from the perspective of evaluation, but also algorithm development and understanding.
>
> We can only speculate as to why the cooperative setting seems to be so dominant within MARL research, but some of the reasons could include: (1) it is a simpler setting and (2) it has more practical appeal from a systems engineering perspective, i.e. most real-world MARL applications seems to be cooperative in nature. However, cooperative MARL still exhibits many challenges specific to the multi-agent setting, including non-stationarity (in the environment from the perspective of each agent, since other agents are also learning), credit assignment (between agents at a particular time step as well as temporally across timesteps), partial observability (in decentralised control settings) and complex coordination. In fact, much of the research leveraging centralised training with decentralised execution (CDTE) in cooperative MARL, are specifically designed to combat these issues. However, we note that ultimately, in the full cooperative setting with shared rewards, although these challenges are present, the evaluation protocols employed mirror much of what is done in the single-agent setting. Therefore, this is also true of our proposed standardised protocol.
>
> * “the domain overfitting issue discussed in Section 2.3 adds a new perspective to the MARL community, but I am unsure whether this perspective alone has sufficient novelty for acceptance”
>
> Given the recent trends shown in our paper of only a few environments being used in the majority of recent publications, and that these specific environments are quite susceptible to manipulation, we find this perspective particularly valuable to the community. The improper use of these few environments could potentially contaminate the majority of new research findings and mislead the field in significant ways. Although we have not proposed concrete solutions to this problem in this work, we feel we have sufficiently been able to expose it and suggest possible solutions that should be investigated in future work.
>
> Finally, we thank the reviewer for pointing out the mistake in Figure 3(f), we have corrected it in the revised version of the paper.

---

> > ### Comment · Reviewer_hmEq · 2022-08-05
> > **Response to Author Rebuttal**
> >
> > I appreciate the authors for their detailed response. I agree that the meta-analyses and the proposed evaluation protocol (along with the release of the resources) are positive, and I have updated my score from 4 to 5.

---

> > > ### Author Response · Authors · 2022-08-08
> > > **Response to Reviewer hmEq**
> > >
> > > We wish to thank the reviewer for going through our response and for updating their score.

---

### Author Response · Authors · 2022-07-29
**Reply to all reviewers**

We would like to kindly thank all the reviewers for their valuable feedback, questions and suggestions. This constructive feedback helped us to improve the quality of the paper. We deeply appreciate the time spent reviewing our paper and the level of interest shown in this work as a potential benefit to the MARL community.

We will address each reviewer’s concerns separately. However, we felt a particular shared concern would be more appropriately addressed in general. This concern, shared amongst several reviewers, can be summarised as follows: many of the issues raised in our work were already highlighted and addressed in work on single-agent RL and that little about the issues we highlight and the suggested standardised protocol is unique to MARL.

We agree with the reviewers. In fact, the structure of the paper illustrates this point clearly by first surveying the single-agent literature and highlighting parallels in MARL. However, we feel that this approach does not detract from our contributions for the following reasons.

The main contributions of our work are (1) our meta-analysis and (2) our proposed evaluation protocol.

In (1), we were able to expose, for the first time, the extent and severity of issues that currently exist within the field of cooperative MARL, highlighting the desperate need for improved standards. This is to say, even if similar issues have existed and been partly addressed in the single agent setting, before this work, it was not clear to what extent the current state of evaluation in MARL was plagued by such similar issues. Before treatment, one must first ascertain the proper diagnosis. Therefore, we feel illuminating the state of current evaluation within the field is already a significant contribution. Furthermore, by publicly releasing our dataset, we invite the community to add to it and contribute additional analyses to help strengthen our understanding of the state of evaluation in MARL and how to further improve it. We see this as a valuable avenue for future research and follow-up contributions.

In (2), our insights from the meta-analysis highlighted that most of the issues raised in current evaluation methodologies ultimately stem from a lack of standardisation and motivated our design and proposal of a standardised evaluation protocol. We already noted in the paper that our innovation regarding the protocol is limited. However, we would argue that standardisation is rarely innovative. In essence, setting standards is choosing the best instantiation of a large set of possible approaches that typically already exist. The primary motivation to set standards is to make comparisons and measuring progress easier and more meaningful. Nevertheless, we would argue that establishing such a standard informed by proper analysis and the study of emerging trends in the field is a significant contribution. We do note that the benefits from a shared protocol are tightly coupled to its adoption by the wider community (i.e. it is only as effective as the degree to which it is being used). Therefore, we feel that having this work published at NeurIPS is perhaps the best venue for aligning the community on how to perform evaluation and achieve the maximum benefit from such a proposal. We hope the reviewers agree.

---

> ### Author Response · Authors · 2022-07-29
> **Resources to be released with the paper to support the community**
>
> To support the community in adopting a common protocol and better understanding of the state of evaluation in cooperative MARL, we are releasing a wide range of resources with our paper including:
>
> * **Evaluation tools**: We believe that if the MARL community can agree on an exact format for raw evaluation data it would make it easier for researchers to process that data and benchmark the quality of their algorithms using standardised tools. Therefore, along with our evaluation protocol, we will also publicly release MARL framework agnostic tools to be used by practitioners to process raw MARL experiment data for downstream use with the suggested evaluation tools provided by the rliable library.
>
> * **Colab notebooks**: We are releasing colab notebooks containing all the code used to perform our meta-analysis as well as examples using our evaluation tools. Our vision is that many more interesting questions can be asked and answered by further analysing the data we collected and therefore we hope that by providing our original notebooks, we can help the community do their own analyses more easily.
>
> * **Reporting templates**: In our protocol, we stress the importance of proper reporting. To assist on this front, we release LaTeX reporting templates that can easily be incorporated by researchers into their papers for future publication.
>
> * **Dataset and tools**: As mentioned above, we are releasing our full dataset, not only in the form of a spreadsheet but as a notion page allowing easy exploration and filtering. Furthermore, we encourage the community to add new data as more papers are published, and for this, we have included guides and tools to make it as easy as possible to contribute to the dataset.
>
> * **Feedback form**: As we mention in the paper, we are realistic in our efforts, knowing that a single protocol is unlikely to be applicable to all MARL research. This includes even settings within cooperative MARL, such as the subfield of cooperative MARL studying communication. Therefore, we think it is extremely valuable to obtain feedback from the community regarding the proposed protocol and how it might be adapted and improved for other settings of interest in MARL.
>
> Finally, we want to highlight that all of the above are (anonymously) linked to on our **paper website**: https://sites.google.com/view/marl-standard-protocol

---

### Meta-Review · Area_Chair_hfKw · 2022-08-26

**Recommendation:** Accept
**Confidence:** Certain

**Metareview:**

This paper performs a meta-research on cooperative MARL research, identifies three main issues, and proposes a standard evaluation protocol. All the reviewers agree that the paper is well written, provides interesting perspectives, and recommends reasonable solutions. A common concern that was shared by all reviewers is that the discussed issues were already highlighted in prior works on single-agent RL. The "Reply to all reviewers" in the rebuttal clearly addressed the above concern. After the discussion, all reviewers agree that publishing this study, including both the meta-analyses and the proposed evaluation protocol, would be beneficial for the MARL community. Thus, I recommend accepting this paper.

**Award:**

No

---

### Decision · Program_Chairs · 2022-09-14

Accept